# A Rapid Self-Supervised Deep-Learning-Based Method for Post-Earthquake Damage Detection Using UAV Data (Case Study: Sarpol-e Zahab, Iran)

**Narges Takhtkeshha [1], Ali Mohammadzadeh [1,\*] and Bahram Salehi [2]**

[1] Department of Photogrammetry and Remote Sensing, K. N. Toosi University of Technology, Tehran 19967-15433, Iran
[2] College of Environmental Science and Forestry, State University of New York, Syracuse, NY 13210, USA
\* Correspondence: a_mohammadzadeh@kntu.ac.ir

**Abstract:** Immediately after an earthquake, rapid disaster management is the main challenge for relevant organizations. While satellite images have been used in the past two decades for building-damage mapping, they have rarely been utilized for the timely damage monitoring required for rescue operations. Unmanned aerial vehicles (UAVs) have recently become very popular due to their agile deployment to sites, super-high spatial resolution, and relatively low operating cost. This paper proposes a novel deep-learning-based method for rapid post-earthquake building damage detection. The method detects damages in four levels and consists of three steps. First, three different feature types—non-deep, deep, and their fusion—are investigated to determine the optimal feature extraction method. A "one-epoch convolutional autoencoder (OECAE)" is used to extract deep features from non-deep features. Then, a rule-based procedure is designed for the automatic selection of the proper training samples required by the classification algorithms in the next step. Finally, seven famous machine learning (ML) algorithms—including support vector machine (SVM), random forest (RF), gradient boosting (GB), extreme gradient boosting (XGB), decision trees (DT), k-nearest neighbors (KNN), and adaBoost (AB)—and a basic deep learning algorithm (i.e., multi-layer perceptron (MLP)) are implemented to obtain building damage maps. The results indicated that auto-training samples are feasible and superior to manual ones, with improved overall accuracy (OA) and kappa coefficient (KC) over 22% and 33%, respectively; SVM (OA = 82% and KC = 74.01%) was the most accurate AI model with a slight advantage over MLP (OA = 82% and KC = 73.98%). Additionally, it was found that the fusion of deep and non-deep features using OECAE could significantly enhance damage-mapping efficiency compared to those using either non-deep features (by an average improvement of 6.75% and 9.78% in OA and KC, respectively) or deep features (improving OA by 7.19% and KC by 10.18% on average) alone.

**Keywords:** damage mapping; self-supervised; deep learning; unmanned aerial vehicle; earthquake management

## 1. Introduction

In the aftermath of a disaster, assessing damage, managing rescue teams, and resettling people all require achieving an understanding of the extent of building destruction in a timely manner [1]. Hence, it is essential to immediately produce building damage maps for affected regions. In light of rapid acquisition and automatic analysis of large areas, remote sensing data has gained great attention in the field of earthquake damage estimation over several decades [2]. Although satellite data has been used for mapping damage in most previous studies [3–9], the very high resolution of UAV data and their ability to generate 3D data, along with ease of access and cost-effectiveness, have prompted researchers to utilize drone data to identify earthquake damage in recent years [10]. Moreover, the

very high spatial resolution of UAV data makes it possible to detect more—and more detailed—damage levels.

Several studies have examined building damage in the aftermath of an earthquake using remote sensing images, and are divided into two and three categories in accordance with event time and method, respectively. An overview of some studies that have used aerial/UAV data can be found in Table 1. A single-temporal approach using only post-event data is more practical and preferable to the multi-temporal approach, given that pre-event images may be lacking, co-registration can be challenging, or geometry may differ [11,12]. Parallel to this, a majority of studies have applied three types of methodology to map building destruction: (1) rule-based decision, (2) machine learning, and (3) deep learning, either object-based image processing (OBIA) or pixel-based. Applying modern deep learning algorithms for classifying damage has two major shortcomings. The first weak point is that these models usually require a significant amount of training on a dataset which are challenging to prepare in time during a disaster [12]. Although deep transfer learning has been applied [13,14] in other studies, the complexity and variety of seismic damage poses a serious risk of poor generality to these models [15]. Further, some studies that have adopted deep object detection methods (e.g., YOLO [16] and SSD [17]) have demonstrated their disadvantages when they come to detecting damage in dense urban areas. On the other hand, machine learning methods require fewer training samples and are faster; however, compared to deep learning methods, they are unable to extract image features automatically. Unlike the two methods discussed—although designing rules for this complicated issue is tricky—rule-based approaches can operate more rapidly and independent of training samples. In addition, the superiority of supervised learning methods over unsupervised ones has been demonstrated by a number of studies. Despite this benefit, their dependence on training samples reduces their generality in practical industrial disaster applications. Consequently, in the past few years, a great deal of attention has been paid to reduce the number of samples that are required to train supervised algorithms. In this domain, self-supervised learning has received special attention in different remote sensing applications [13–15,18–20]. In the case of disaster management, Ghaffarian et al. [20]—by automatic selection of training samples based on pre-event OpenStreetMap building data—updated the required building database for damage detection.

When it comes to responding to an emergency, time, accuracy, and cost are three crucial elements that cannot be overlooked. Nevertheless, the processing time of damage mapping has been little reported in research up to now. To date, a number of different standards have been introduced for the level of damage arising from an earthquake [21–26]. Among them, EMS-98 [21] is one of the widely accepted damage standards which, in five degrees, reflects the ideal detectable damage levels in situ by an agent. Having said this, some studies have involving only two classes, damaged and undamaged (see Table 1). Accordingly, paying substantial attention to damage degrees is important for more accurate destruction mapping. On the other side, the limitation of earth observation in capturing all the information necessary to determine inherently complex building damage makes it essential to define the indicated damage degrees in well-known standards based on remote sensing data [10,26]. This is what we called the adaptation of the famous EMS-98 standard [21] to remote sensing data in this study. Moreover, the amount of automation not only leads to a decline in processing time and energy consumption but also has a significant impact on cost reduction. As such, fully automated solutions in rescue operations are definitely highly preferred. More importantly, as indicated in [10,27], damage mapping in two of the foremost emergency management services (i.e., EU Copernicus [28] and Charter [29]) are still limited to visual interpretation of high-resolution remote sensing images.

For these reasons, this study was primarily aimed at assessing the destruction that occurred after the Sarpol-e Zahab earthquake by striking a balance between the three aforementioned crisis management factors as much as possible. Particularly, the main contributions of this study are as follows.

1.      Adaptation of the EMS-98 standard's damage levels to very-high-resolution UAV images.

2.   Proposing a novel method for automatic selection of reliable training data needed for consecutive supervised damage detection.
3.   Presenting a rapid and local-normalized digital surface model (nDSM) for expediting the damage mapping procedure.
4.   Investigating various deep and non-deep features and the fusion of them in the framework of eight machine/deep learning algorithms.

In the remaining sections, the proposed framework is introduced in Section 3. Next, experimental results and discussion are reported in Section 4. Finally, Section 5 concludes this article as well as provide outlooks of this research domain.

**Table 1.** Studies that have used UAV/aerial imagery to assess earthquake-induced damage, listed by date.

| Damage Level(s) | Methodology | Data-Resolution (m) | Study |
|---|---|---|---|
| Undamaged, minor damage, and collapsed | Rule-based—OBIA | Arial—0.5 | [30] |
| Intact, damage, and collapsed | Machine learning—OBIA | UAV—0.5 | [31] |
| Damaged and undamaged | Machine learning—OBIA | Arial—0.1 | [32] |
| Damaged and undamaged | Rule-based—OBIA | UAV—0.2/Arial—0.61 | [11] |
| Intact, broken, and debris | Deep learning (CNN) | Arial—0.67 | [33] |
| Intact, partially collapsed, and collapsed | Deep learning (Deeplab v2) | Arial—0.5 | [6] |
| Undamaged and debris | Deep learning (SSD) | Arial—0.3 | [17] |
| Undamaged, minor damage, and debris | Deep learning (CNN) | Arial—0.25 | [7] |
| 0.025 m: Basically intact, slight damage, partially collapsed, completely collapsed 0.079 m: Basically intact, partially collapsed, completely collapsed | Rule-based | UAV (orthophoto+ point cloud)—0.025 and 0.079 | [34] |
| Slight damage, moderate damage, and serious damage | Deep learning (Inception V3) | Arial—0.3 | [35] |
| Collapsed | Deep learning (YOLO V3) | Arial—0.5 | [16] |
| Damaged and undamaged | Deep learning (ADGAN) | UAV—0.02 and 0.06 | [36] |
| A wide range of minor to major damage | Deep learning (CNN) | UAV—0.09 | [37] |
| Collapsed | Faster R (CNN) | UAV—0.1 and 0.15 | [38] |

## 2. Study Area and Dataset

The region of Sarpol-e Zahab in Kermanshah, Iran, was struck by an earthquake on 12 November 2017 with an intensity of 7.3 Mw. Because of this event, over 620 people were killed and 7000 people injured. Furthermore, nearly 70,000 people lost their homes [39]. The location map of the study area, UAV-derived orthophoto and DSM, and the produced reference building damage map encompassing previously mentioned damage levels $L_1$–$L_4$ are shown in Figure 1. In order to detect the earthquake-induced damage levels, about 250 UAV images were processed and two orthophoto images with ground resolution of 2.5 cm and 25 cm were produced. A similar action was carried out to produce digital surface models (DSMs) with spatial resolution of 5 cm and 25 cm. The achieved orthophoto and DSM with resolution of 25 cm were used as input of the proposed method while the higher resolutions ones are used by an expert in reference map production. Further, buildings' footprint shapefiles were prepared from a manual delineation of the corresponding pre-event Google Earth images due to the lack of proper and updated vector data from the Sarpol-e Zahab region. Moreover, Table 2 shows descriptions of the acquired UAV data.

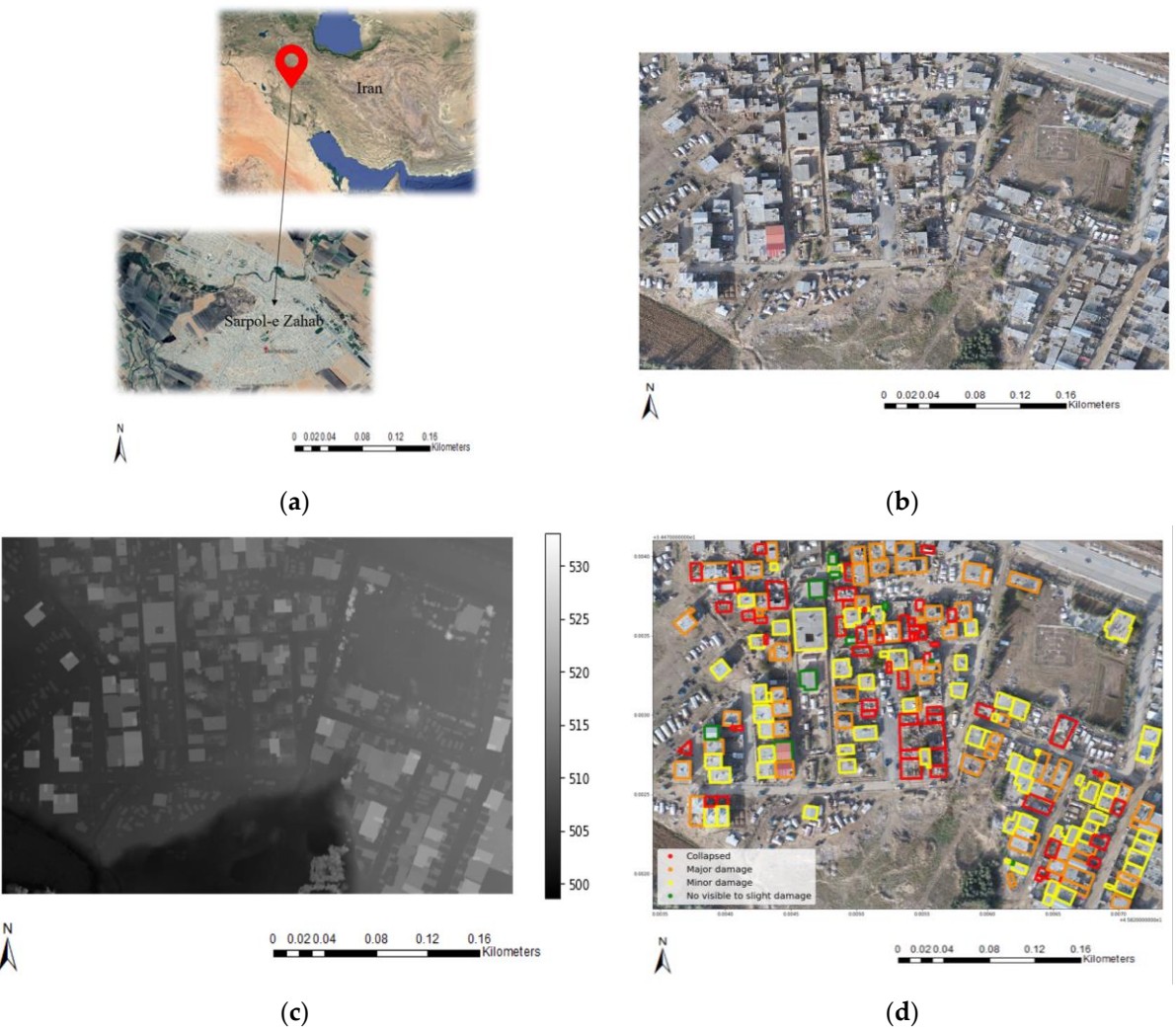

**Figure 1.** Study area: (**a**) Location map, (**b**) Orthophoto, (**c**) DSM, (**d**) Buildings' vector map and the corresponding reference damage levels.

**Table 2.** Utilized UAV dataset specification.

| UAV Device | Phantom 4 Pro |
| --- | --- |
| Flight altitude | 98.8 m |
| Camera | FC6310 |
| Focal length | 8.8 mm |
| Image dimension | $3648 * 5472$ pixel |
| Pixel size | 2.41 μm |

## 3. Methods

According to our adaptation of the EMS-98 standard to very-high-resolution UAV images, four damage levels are considered in this study, including "no visible to slight damage ($L_1$)", "minor damage ($L_2$)", "major damage ($L_3$)" and "collapsed ($L_4$)". In our adapted damage scale, $L_1$ and $L_2$ refer to the two first grades among the five indicated damage grades in EMS-98. The grade $L_4$ is the last grade in EMS-98, as well. In fact, $L_3$ is the combination of "substantial to heavy damage (grade 3)" and "very heavy damage (grade 4)" in EMS-98. Further information about this adaptation is discussed in Section 3.2. The proposed method for self-supervised building damage mapping comprises three main steps. The general framework of the presented damage detection approach is depicted in Figure 2.

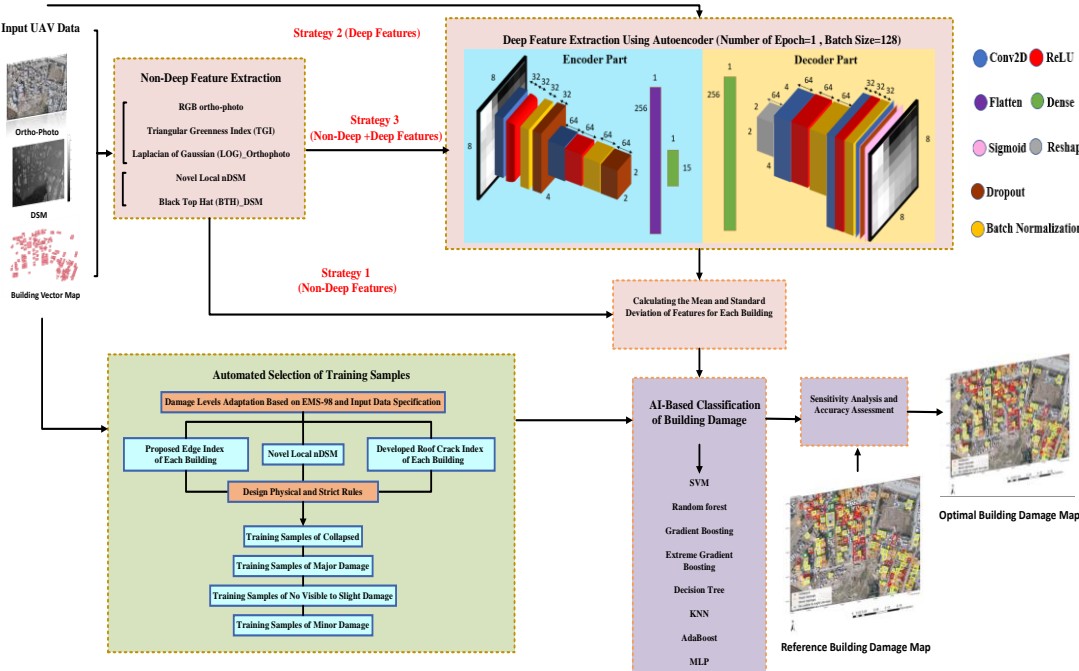

**Figure 2.** The general framework proposed for producing the damage map of buildings.

In the first step, three different feature-extraction strategies are carried out to investigate their suitability for optimal detection of damage levels. These strategies embrace extracting (1) non-deep features (NDF), (2) deep features (DF), and (3) fused deep and non-deep features (FF). In the first strategy, in addition to the RGB bands of the orthophoto, triangular greenness index (TGI) and Laplacian of Gaussian (LOG) are calculated from those bands. Moreover, a novel local nDSM and black top hat (BTH) are calculated from input DSM data. In strategy 2, a one-epoch convolutional autoencoder (OECAE) is designed to extract deep features from a stacked form of the orthophoto and DSM. In the last feature-extraction strategy, non-deep features are fed to the OECAE to achieve fused deep and non-deep features. Subsequently, mean and standard deviation (SD) of the extracted features for each feature-extraction strategy are calculated and fed to the classification step as the final features. In the second step, a fully automatic procedure based on physical and strict rules is proposed to select training samples of all the building damage levels. In this step, a novel roof crack index and edge index are presented to facilitate designing of efficient rules for selection of the training samples. Following that, in order to achieve damage mapping, some AI-based classifiers—including seven machine learning methods and a basic deep learning method—are implemented in the final step. Utilizing each of the three mentioned feature-extraction strategies in each of the eight classifiers resulted in a total of 24 damage maps. Eventually, the final map is made by conducting a comprehensive analysis of these 24 maps. Detailed explanations of individual steps are given in the following sections.

### 3.1. Feature-Extraction Strategies

In the first step, three different strategies are designed and implemented which use unique and novel features. The list of implemented features is presented in Table 3. In strategy 1 (NDF), RGB bands of the input orthophoto are directly considered as three non-deep features. Additionally, according to Equations (1) and (2), TGI and BTH are obtained from input orthophoto and DSM, respectively. In Equation (2), 'I' and 's' refer, respectively, to input DSM and morphological structuring element [40–42]. BTH is based on closing morphological operations and highlights damaged areas which have instinctively lower height than their adjacent neighborhood regions. Since a grayscale image is usually

needed to extract textural features, we used Equation (3) [41] to create a panchromatic band from our RGB image before computing LOG of the orthophoto. LOG is one of the most robust methodologies which detects edges by identifying zero-crossing through applying the Laplacian filter after the Gaussian filter [42]. For an image f(x, y), the final equation of LOG can be found in Equation (4), where $\sigma$ is the standard deviation of the Gaussian function. Equation (5) defines how the nDSM feature can be calculated by subtracting DTM from DSM, in which DTM is bare earth of the study area. Various filtering algorithms such as Sohn filtering [43] and Axelsson filtering [44] can be implemented to obtain DTM from DSM. In this study, alternatively, instead of using DTM for nDSM calculation, a novel local nDSM is proposed to dedicatedly better highlight local and relative height differences in damaged regions.

$$TGI= -0.5 \, (190 \, (Red - Green) - 120 \, (Red - Blue)) \tag{1}$$

$$BTH(I) = closing \, (I, s) - I \tag{2}$$

$$Pan = 0.2989 \, Red + 0.587 \, Green + 0.114 \, Blue \tag{3}$$

$$LoG(x, \, y) = \frac{-1}{\pi\sigma^4} \left[ 1 - \frac{x^2 + y^2}{2\sigma^2} \right] e^{\left(-\frac{x^2+y^2}{2\sigma^2}\right)} \tag{4}$$

$$nDSM = DSM - DTM \tag{5}$$

**Table 3.** Applied features in each strategy.

| Utilized Features | Features Type | Strategy No. |
| --- | --- | --- |
| RGB bands of orthophoto, TGI, BTH_DSM, LOG_orthophoto, novel local nDSM | Non-deep features (NDF) | 1 |
| OECAE-based deep features extracted from stacked input orthophoto and DSM | Deep features (DF) | 2 |
| OECAE-based deep features extracted from non-deep features of strategy 1 | Fused non-deep/deep features (FF) | 3 |

Figure 3 shows the flowchart of the proposed novel local nDSM algorithm. In this regard, it should be noted that, due to fact that only building damages is our objective, we simply developed a fast method to calculate nDSM in these areas. For each building from the vector map, a buffer zone is created. The associated buffering distance is equal to the size of the average road width surrounding the corresponding building which, in our study area, is about 10 m. Following this, we masked DSM with these buffers and then computed the minimum value of DSM in each building. Eventually, in each building, for building interior pixels, nDSM values are calculated by subtracting minimum values of DSM pixels from DSM values. Consequently, we calculated nDSM locally by involving the DSM values in buffer polygons around each building. The dominant advantage of our local nDSM over the traditional way is preventing negative nDSM values, as well as not requiring to calculation of DTM. Thus, the proposed local nDSM is expected to be much faster than the existing time-consuming nDSM production process. As a result, the proposed novel nDSM would lead to achieving more practical and operational emergency responses in crisis management. In Section 4.1, we discuss the local nDSM in detail and compare it with that obtained from the LiDAR extension of ENVI software.

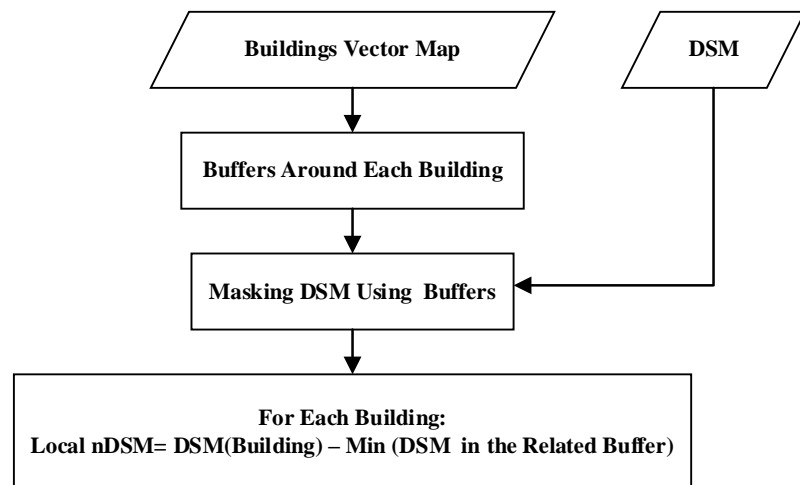

**Figure 3.** The flowchart of the proposed local nDSM.

Contrary to non-deep features, the emergence of state-of-the-art deep neural networks has made it possible to extract deep features (DF) automatically and more directly [45–49]. Autoencoders (AEs) are common unsupervised networks in deep learning algorithms that basically compress the input data into a relatively lossy latent-space representation automatically (encode part), then reconstruct the input from this representation through the decode part [48]. In the literature, autoencoders have been used in various applications in the field of remote sensing imagery. For instance, AE is used for speckle denoising of synthetic aperture radar (SAR) data [50–52], target recognition [53,54], change detection [55–57], and as a non-linear dimensionality reduction technique in hyperspectral image processing [58,59]. In recent studies, AE has received attention as an automatic feature extractor, without any labeled training data for clustering/classification tasks required [58]. Actually, in this application, the decoder section of AE is ignored. That is why we designed a convolutional autoencoder to extract deep features in the second strategy (DF) from the stacked orthophoto and DSM. In addition, we should point out here that in order to speed up the feature-extraction procedure, the number of epochs in our CAE has been set equal to just one. Moreover, in the third feature-extraction strategy (FF), as the potential of fusing non-deep and deep features in accuracy improvement has been confirmed in a number of recent studies [60–62], we fed non-deep features introduced in strategy 1 through the same AE network used in strategy 2 to get fused non-deep/deep features.

*3.2. Automated Selection of Training Samples*

In order to automatically select the required training samples in the four aforementioned damage degrees, a set of strict and simple rules was designed based on the adaptation of UAV data with the EMS-98 standard (see Table 2). First of all, since pancake buildings are low in average height by nature and, on the other hand, have heaps of debris which lead to considerable edges on the orthophoto, we selected training samples of the "collapsed" class (including pancake and heap of debris) by applying a 3 m threshold to the median of the presented local nDSM as well as a 90% threshold to the proposed edge index. As depicted in Figure 4a, our presented edge index is based on applying the canny edge detection operator to orthophoto in conjunction with the mean shift filter and morphological erosion to detect more strong edges. Furthermore, since edge pixels have a skeleton form, the OpenCV module in Python was used to highlight them. Ultimately, the percentage of edge pixels in each building was regarded as the edge index. After that, as anomaly in height can frequently result from noticeable damage, the remaining buildings with a standard deviation of nDSM above 0.3 were confidently considered as having "major damage". Further, picking the training samples of the destruction map becomes more

challenging when it comes to the "minor damage" class as well as the "no visible/slight damage" for the following reasons:

- Minor damage to buildings can predominantly be seen as a noticeable amount of roof surface cracking. Based on our examination, these cracks do not result in tangible differences in height. In light of this, optical data should be incorporated since elevation information of DSM is not appropriate for distinguishing $L_2$ damage levels from $L_1$.

- Buildings in Iran often have cracks appearing in brown spots due to roof elements peeling away, commonly bituminous waterproof layers. The important point to be noted here is that the resemblance of shadows and crack spots in terms of having low orthophoto digital number value results in identifying shadows besides cracks in most crack-detection algorithms. Therefore, because of the mentioned fact, an actually intact building which has shadows in the image according to the illumination conditions can be identified as crack spots by mistake. Hence, the shade of a building acts as a destructive factor in determining minor damage to buildings.

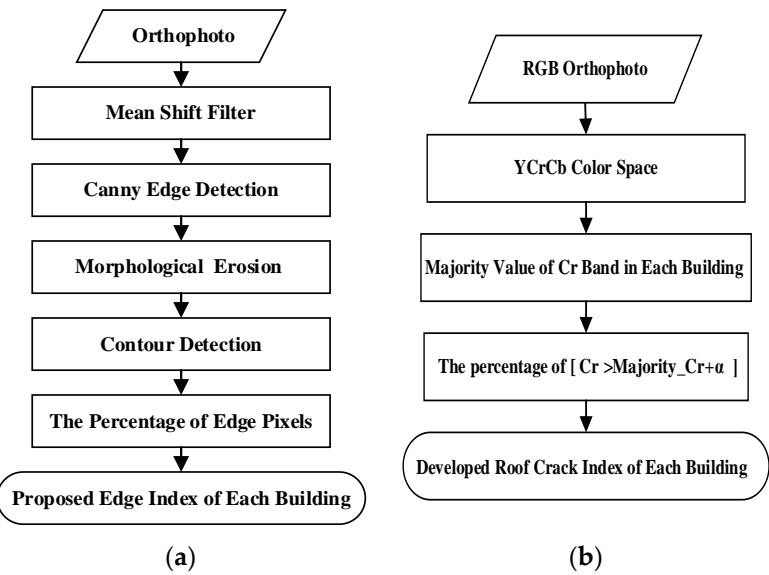

**Figure 4.** (**a**) The flowchart of the presented edge index, (**b**) the flowchart of the developed roof crack index.

To address the problems outlined above, we put forward a novel crack index that mitigates the stated negative effect of shadow. For this purpose, according to our experiments, by converting the RGB color space to the YCrCb one, cracks and shadows respectively become lighter and darker than the majority value in the Cr band. Thus, this color space transformation, by creating differences between shadows and crack spots, helps in identifying the real cracks. So, the crack index is calculated by first converting RGB to YCrCb and then calculating the percentage of pixels with values above the dominant value in the Cr band plus a threshold ($\alpha$) for each building (Figure 4b). Because our UAV image is an eight-bit integer, $\alpha$ is taken as equal to four, which differs only a little from the majority value. Consequently, using this novel crack index, pixels that have a negligible height difference—i.e., their normalized SD of nDSM—between 0 and 1—is less than 0.05—but have a perceptible amount of crack index (10%) are selected as training samples for $L_2$ damage class, while buildings with insignificant crack index values (less than 5%) are chosen for the $L_1$ damage degree. Table 4 summarizes the previously mentioned rules for auto-training samples.

**Table 4.** Rules for automated selection of training samples.

| Selection Rule(s) | Damage Level (in Order) |
|---|---|
| Median(nDSM) < 3 and Edge Index > 90% | Collapsed |
| Normalized_SD (nDSM) $\geq$ 0.3 | Major Damage |
| Normalized_SD (nDSM) < 0.05 and Crack Index < 5% | No Visible to Slight Damage |
| Normalized_SD (nDSM) < 0.05 and Crack Index > 10% | Minor Damage |

### 3.3. AI-Based Classification of Building Damage

In this research, seven machine learning methods as well as a basic deep learning method have been used for classification, which are as follows: (1) SVM, (2) random forest, (3) gradient boosting, (4) extreme gradient boosting, (5) decision tree, (6) k-nearest neighbors, (7) adaBoost, and (8) a four-layer MLP. The multilayer perceptron is a basic deep learning network, and the rest of them are machine learning algorithms. Additionally, among the mentioned machine learning methods, decision trees, random forests, gradient boosting, extreme gradient boosting, and adaBoost are kinds of ensemble learning. After extracting features as well as an automated selection of the training samples, the mentioned machine/deep learning methods were used to generate the building damage maps. It is worth mentioning that, for the sake of speeding up the generation of damage map, only the mean and standard deviation of calculated features in each building polygon were involved in classification and obtaining of damage level maps.

### 4. Results and Discussion

The implementation of proposed damage detection method includes three main steps: feature extraction, automatic training sample selection, and optimal damage map production. In the first step, the mentioned features are produced, among which the proposed local nDSM plays a key role. Therefore, its performance and comparison with a widely used nDSM based on conducting point cloud filtering in ENVI LiDAR software is presented and discussed. Afterwards, performance of automatic training sample selection is evaluated in the second part along with a discussion of the effect of the proposed local nDSM on it. The last subsection of this part comprises the resultant damage maps, followed by assessing the accuracy of them. The produced damage maps were validated using popular accuracy evaluation criteria derived from the confusion matrix, including overall accuracy, kappa coefficient, user accuracy (UA), and producer accuracy (PA). Moreover, evaluation of damage mapping in terms of feature-extraction strategies (NDF, DF, and FF) and type of classification algorithm is also presented in this step. Furthermore, since this study relies on automated training samples, a comparison of the accuracy of these automated training samples against manually selected ones is presented at the end of this section. All of the implementations were accomplished using the Python language and TensorFlow framework on a system with a Core (TM) i7-10870H CPU running at 2.20 GHz and 16GB RAM.

### 4.1. Features Production

With the aim of evaluating the impact of the feature extraction method on damage mapping, required features for classification were extracted using three strategies embracing non-deep features (NDF), deep features (DF), and fused non-deep/deep features (FF). During NFD extraction, a disk kernel with a 7-pixel radius was used to calculate BTH of DSM. For LOG computing, a 3-by-3 window size was used for the Gaussian filter. For local nDSM, we considered buffer size equal to 10 m based on the average road width of our study area. Additionally, we evaluated the time efficiency and accuracy of our proposed local nDSM against nDSM derived in ENVI LiDAR software, version 5.3. For this purpose, we produced the DTM by first filtering sparse point clouds of the study area (density = 94.49 points per square meter), and then gridding it in Global Mapper software using the "DTM" gridding method. Subsequently, nDSM was produced according to (5).

We applied "Urban Area Filtering" as well as "Height at Average Roof" modes for filtering UAV-derived point clouds in ENVI LiDAR, while other parameters were left as default. Our empirical results indicated that nDSM generation based on ENVI LiDAR takes 65.7 s, whereas the proposed local nDSM produces it in only 5.6 s. As such, our method has been able to speed up nDSM production by 91.47%. Figure 5 compares these two nDSM production strategies, demonstrating the superiority of our local nDSM over ENVI. It can be seen that ENVI nDSM mostly deformed height changes in some areas and mostly removed debris areas, whereas local nDSM highlights debris and has a similar elevation form to DSM. Moreover, despite the fact we generally expect negligible height variance for intact buildings, ENVI nDSM caused elevation difference for some "no damage" buildings such as zoomed building 2 in Figure 5c. According to Figure 5c, in comparison with ENVI nDSM, the proposed local nDSM normalizes elevation more closely to DSM and yields a more reliable elevation difference in all damage levels. Hence, from the perspective of being both time-efficient and highly accurate, the novel local DSM can have an enormous impact on damage mapping. Therefore, it can be deduced that the proposed local nDSM is well-suited to produce an efficient height feature in order to enhance the damage detection process.

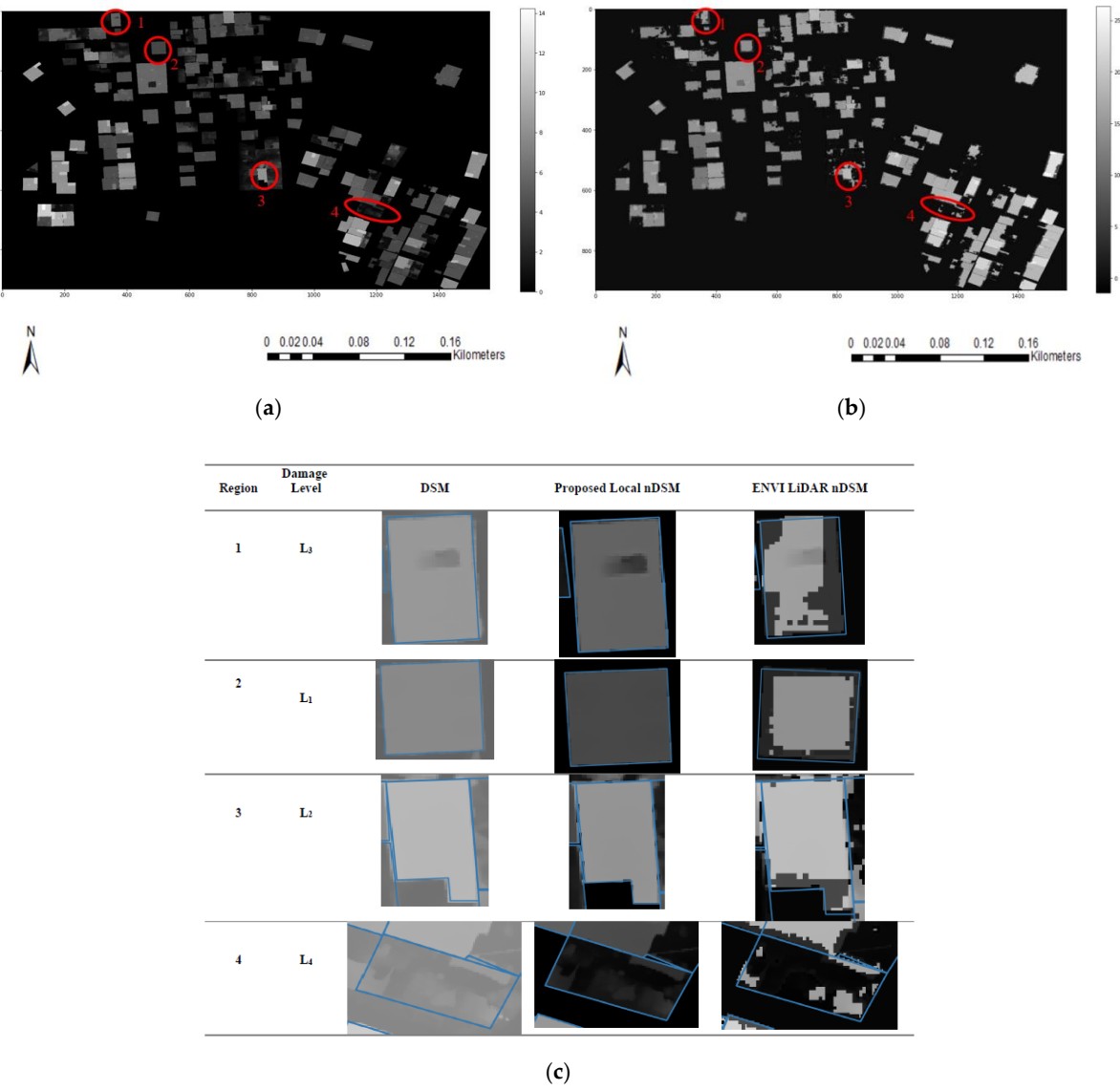

**Figure 5.** Comparison of the accuracy of nDSM: (**a**) Proposed local nDSMs, (**b**) ENVI LiDAR nDSM, (**c**) Zoomed regions over different damage levels.

In addition, according to Section 3.1, we designed a one-epoch autoencoder network for DF and FF feature-extraction strategies. Utilized hyperparameters can be found in Table 5.

**Table 5.** The set hyperparameters for the designed convolutional autoencoder in feature extraction part.

| Hyperparameter | Value |
|---|---|
| Optimizer | Adam |
| Loss | Mean Squared Error (MSE) |
| Learning rate | 0.001 |
| Number of epochs | 1 |
| Batch size | 128 |

The detailed structure of OECAE is represented in Table 6, in which N refers to the number of images that the stacked form of which are fed into CAE. According to Table 3, N is equal to 4 and 7 for DF and FF, respectively. In addition, the batch size hyperparameter is considered to be 128. In this network, Batch Normalization (BN) layers are employed to accelerate the process of network learning [61]. Additionally, dropout regularization layers with a value of 30% are applied which, by ignoring random connections, reduces overfitting and promotes the generalization ability of our OECAE [62].

**Table 6.** The configuration of the designed convolutional autoencoder for feature extraction.

| Block | Unit | Input Shape | Kernel Size | Output Shape |
|---|---|---|---|---|
| Encoder | Conv2D+ReLU+BN+Droput | $8 \times 8 \times N$ | $3 \times 3$ | $4 \times 4 \times 32$ |
| | Conv2D+ReLU+BN+Droput | $4 \times 4 \times 32$ | $3 \times 3$ | $2 \times 2 \times 64$ |
| | Flatten | $2 \times 2 \times 64$ | $2 \times 2$ | $1 \times 1 \times 256$ |
| | Dense | $1 \times 1 \times 256$ | $1 \times 1$ | $1 \times 1 \times 15$ |
| | Dense | $1 \times 1 \times 15$ | $1 \times 1$ | $1 \times 1 \times 256$ |
| | Reshape | $1 \times 1 \times 256$ | $2 \times 2$ | $2 \times 2 \times 64$ |
| Decoder | Conv2D+ReLU+BN | $2 \times 2 \times 64$ | $3 \times 3$ | $4 \times 4 \times 64$ |
| | Conv2D+ReLU+BN | $4 \times 4 \times 64$ | $3 \times 3$ | $8 \times 8 \times 32$ |
| | Conv2+Dropout+Sigmoid | $8 \times 8 \times 32$ | $1 \times 1$ | $8 \times 8 \times N$ |

*4.2. Implementation of Automatic Training Samples Selection*

Figure 6a shows the automated selected training samples in four damage levels derived by using local nDSM and the designed strict rules according to Table 4. The chosen training samples consist of 101 buildings out of 200. The accuracy of selected training samples is evaluated regarding the prepared ground truth of damage levels (Figure 1d). Figure 6c, reveals that the combined use of designed simple rules and local nDSM results in a satisfactory 93.07% overall accuracy and 88.61% kappa coefficient in selecting training samples automatically. In other words, the proposed strict rules have been able to detect the damage level of about half of the buildings with high accuracy. Notably, we will use these buildings as training samples to identify the damage class of the rest buildings, which are in the "none" class in Figure 6a, by utilizing AI-based classification algorithms. In fact, instead of pixel-level labels, our labels are polygon-level, which are faster at processing and easier to obtain, but may lose some information as a result. In addition, in terms of class accuracy, the opted strategy has reached 100% user accuracy for both "minor damage" and "major damage". In addition, the produced accuracies of two other damages of "no visible to slight damage" and "collapsed damage" are 100%. So, $L_4$ and $L_1$ are, in order, the most and the least accurate damage levels which are selected automatically. The fact is that "collapsed" buildings are the most distinguishable and easily recognizable due to their distinct characteristics—either less median nDSM or heap of debris. On the other hand, as discussed in Section 3.2, detection of "no visible damage level" is challenging due to its

nature. On top of these, as nDSM-based elevation information—median and SD—were majorly applied in training sample selection, we examined the effect of novel local nDSM on it against to ENVI nDSM. Broadly speaking, with regard to Figure 6b,c, selected training samples based on local nDSM were by far more accurate than ENVI nDSM. Indeed, using ENVI LiDAR-derived nDSM in place of proposed local nDSM in this application could result in a substantial drop in overall accuracy of 34.51% as well as a sharp decrease in kappa coefficient by 19.62%. More surprisingly, no training samples for the "minor damage" level is chosen when applying ENVI nDSM, as can be seen in Figure 6b,c.

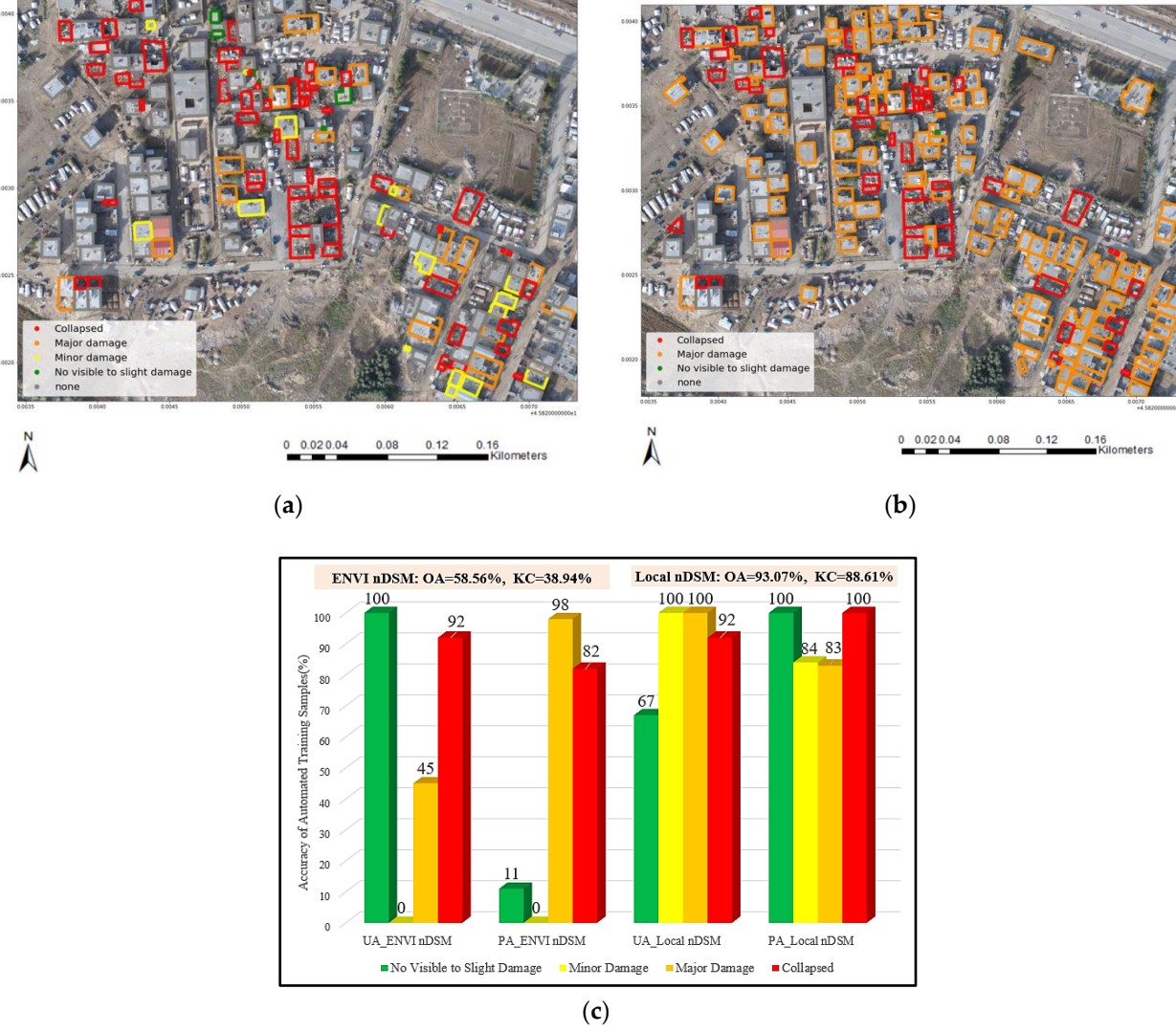

**Figure 6.** Automated selection of training samples: (**a**) detected training samples with applying novel local nDSM, (**b**) selected training samples with applying ENVI LiDAR nDSM, (**c**) accuracy assessment.

### 4.3. Damage Mapping and Evaluation

After features extraction and preparing training samples automatically and with the assistance of local nDSM, we examined eight AI-based learning algorithms on damage level detection. Table 7 indicates the hyperparameters used for each classification algorithm. It is worth noting that these parameters are set experimentally, and also that other parameters are left as are by default of the Sklearn library in Python. Consequently, 28 damage maps in four levels, $L_1$–$L_4$, were achieved, of which the optimal and the worst per three feature strategies are illustrated in Figure 7. Further, the accuracy of the produced building damage maps based on four reputable criteria, including OA, KC, UA, and PA, are represented in Figure 8.

**Table 7.** The set hyperparameters for each learning method.

| Learning Algorithm | Hyperparameters |
|---|---|
| SVM | C = 100, kernel = 'poly', degree = 2, gamma = 'auto', coef0 = 0.1, random_state = 0 |
| RF | n_estimators = 200,max_depth = 10, random_state = 0 |
| GB | n_estimators = 5000, learning_rate = 0.1, max_depth = 1, random_state = 0 |
| XGB | n_estimators = 5000, max_depth = 10 |
| DT | max_depth = 2, random_state = 0, min_samples_leaf = 5 |
| KNN | n_neighbors = 2 |
| AB | n_estimators = 5000, random_state = 0, learning_rate = 0.1 |
| MPL | hidden_layer_sizes = (20,15), random_state = 0, verbose = True, learning_rate_init = 0.03, max_iter = 5000 |

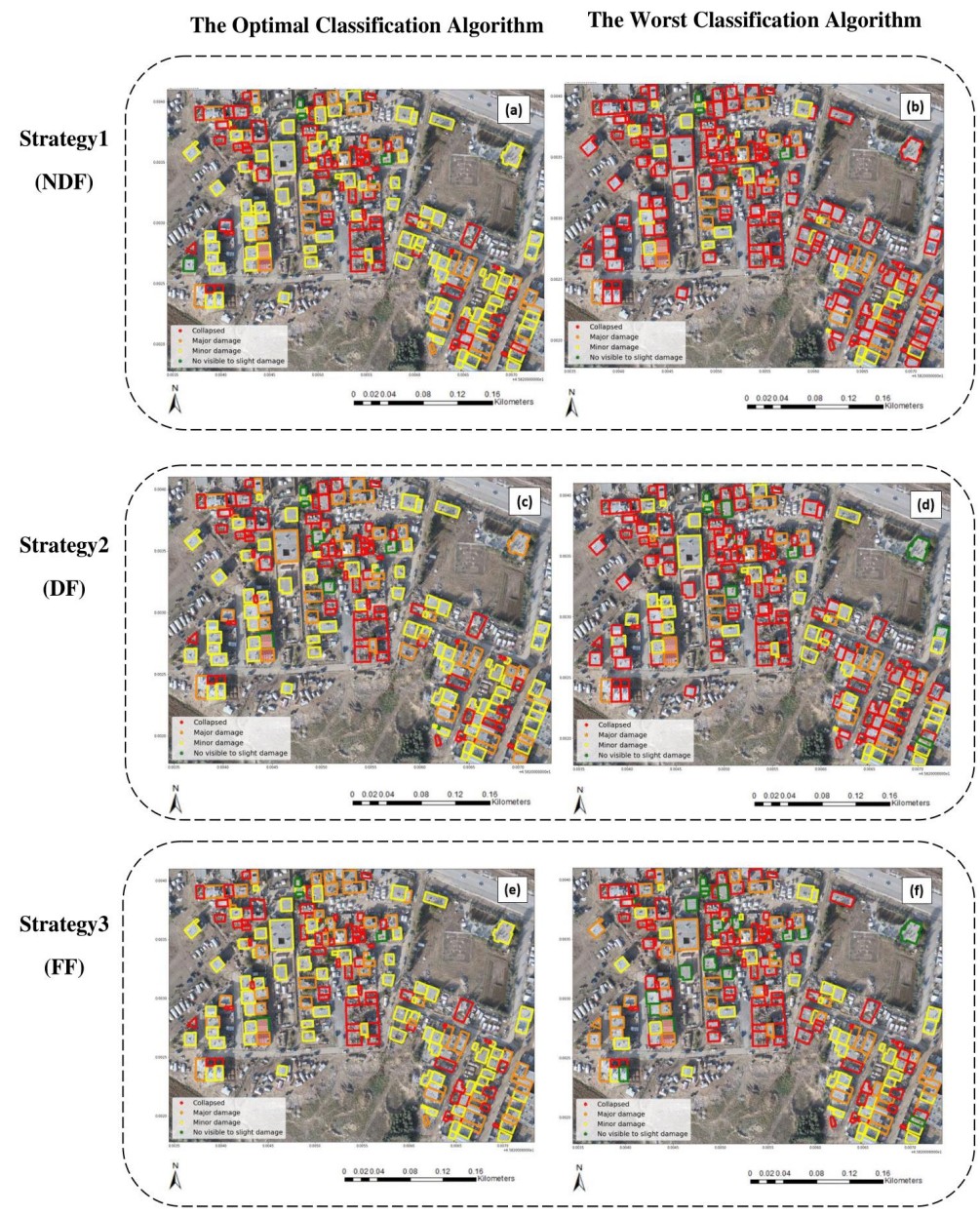

**Figure 7.** The best and worst buildings' damage maps using three different feature strategies and eight AI-based classification algorithms: (**a**) GB (NDF), (**b**) AB (NDF), (**c**) MLP (DF), (**d**) AB (DF), (**e**) SVM (FF), (**f**) AB (FF).

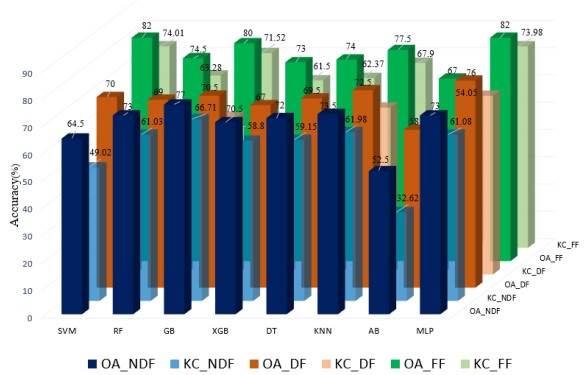

(**a**)

| Classification | Ovrall Accuracy (OA) | | | Kappa Coefficient (KC) | | |
|---|---|---|---|---|---|---|
| Algorithm | OA_NDF | OA_DF | OA_FF | KC_NDF | KC_DF | KC_FF |
| SVM | 64.5 | 70 | 82 | 49.02 | 57.12 | 74.01 |
| RF | 73 | 69 | 74.5 | 61.03 | 55.61 | 63.28 |
| GB | 77 | 70.5 | 80 | 66.71 | 57.91 | 71.52 |
| XGB | 70.5 | 67 | 73 | 58.8 | 52.95 | 61.5 |
| DT | 72 | 69.5 | 74 | 59.15 | 56.21 | 62.37 |
| KNN | 73.5 | 72.5 | 77.5 | 61.98 | 61.22 | 67.9 |
| AB | 52.5 | 58 | 67 | 32.62 | 40.53 | 54.05 |
| MLP | 73 | 76 | 82 | 61.08 | 65.6 | 73.98 |

(**b**)

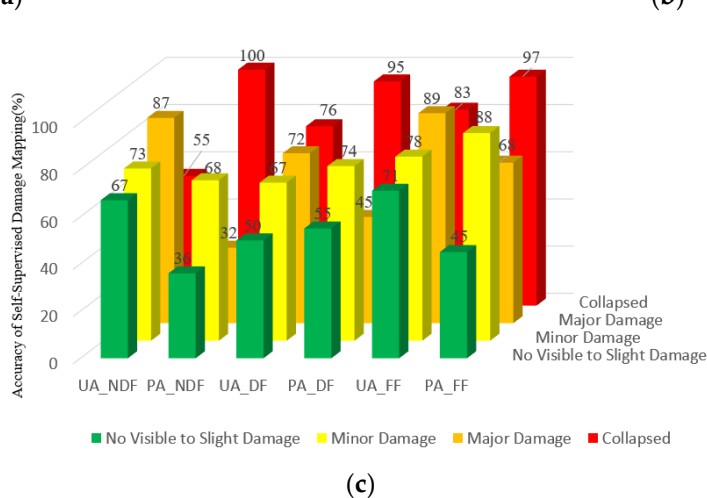

(**c**)

**Figure 8.** Accuracy assessment of three different feature-extraction approaches along with various types of classification algorithms: (**a**,**b**) overall accuracy and kappa coefficient, (**c**) user and producer accuracy per damage level for SVM algorithm (the optimal AI method in this case).

Based on the Mean Squared Error (MSE- (6)) loss function, the loss of designed CAE reached 0.02 in both DF and FF feature-extraction strategies. In Equation (6), $n$: number of data points, $y_i$: observed values, and $\hat{y}_i$: predicted values.

$$\text{MSE} = \frac{1}{n}\sum_{i=1}^{n}(y_i - \hat{y}_i)^2 \tag{6}$$

Darker graphs in Figure 8a refer to the overall accuracy and lighter ones are connected to the kappa coefficient. Figure 8a,b vividly demonstrates the higher OA and KC of fused non-deep/deep features (FF) over the two other feature-obtaining strategies for all eight classification methods. In other words, in comparison to NDF, FF is able to produce more accurate damage maps of buildings with an average improvement of 6.75% and 9.78% in OA and KC, respectively. On top of this, feature fusion has improved the OA and KC of damage level detection by, respectively, 7.19% and 10.18 % on average compared to using non-deep features alone. So, combining deep and non-deep features could strengthen the damage mapping accuracy more rather than specifically and merely employing pure deep features. In spite of this result, deep and non-deep features both show unstable patterns of accuracy improvement over AI-based predictors. Another striking finding is that the performance of adaBoost was the lowest across all three scenarios of feature extraction. Over and above that, SVM and MLP both exhibited the highest overall accuracy when fusing non-deep and deep features (82%). Additionally, their kappa coefficients were too close, 74.01% for SVM and 73.98% for the other. Hence, among eight classification algorithms, SVM is the most accurate method with a slight superiority over MLP. This implies that

a simpler machine learning algorithm such as SVM, in this case, can correspond to a more sophisticated deep learning classifier, of which MLP is one of its basic methods. Moreover, since the SVM algorithm achieved the most accurate damage map, we calculated both the user's accuracy and the producer's accuracy merely for three damage maps associated with this learning method to determine the effect of the feature-extraction strategy on detecting each level of damage. So, according to Figure 8c, the general trend of UA and PA—except for producer accuracy of $L_1$ and $L_4$—proves the dramatic damage class accuracy escalation that is achieved by fusing non-deep and deep features (FF) against NDF and DF feature-extraction strategies.

Additionally, similar to what was concluded and examined in Section 3.2, in all three feature-extraction strategies, $L_4$ and $L_1$ had the most and the least user and producer accuracy. In addition, taking into account all results prove that merely employing the mean and SD of each feature's value in lieu of engaging all of them can result in hopeful accuracy in addition to quicker processing. Aside from that, the efficiency of setting just one epoch for deep feature extracting (DF and FF strategies) can be concluded.

Elapsed time for the self-supervised damage mapping methodology put forward can be found in Table 8. As this table shows, while extracting non-deep features only takes only up to 10 s, deep features and fused features require more time. Therefore, regarding the FF as well as SVM which are, respectively, the most accurate feature strategy and classification method based on experimental results, the proposed damage level detection framework takes 4 min and 20.6 s in total. As this time is related to CPU processing by a normal laptop with the previously stated characteristics, the processing time of the developed damage mapping method could be reduced by GPU processing. Furthermore, regardless of feature-extraction strategy, the automatic selection of training samples plus the classification of damage degrees accounts for only 18.6 s of the damage mapping procedure. This result arises from simplicity of the designed strict selection rules for training samples as well as utilizing only mean and SD of features in each building polygon.

**Table 8.** Processing time of proposed self-supervised method for mapping building damage for each feature-extraction strategy.

| Automated Training Samples Selection and Classification | Feature-Extraction Strategy | | |
|---|---|---|---|
| 18.6 s | FF = 4 min 2 s | DF = 2 min 55 s | NDF = 9.8 s |

### 4.4. Automatic vs. Manual Training Samples

Since we selected all the required training samples with any supervision, the proposed approach has a high level of automation. In this regard, it is worth exploring the impact of the type of training sample. Considering this, we compared the efficiency of the automated training samples with manual ones. As the combination of FF feature extraction and SVM gained the optimal damage map (see Figure 8a,b), we used this approach and considered four different train/test ratios of manual training data, including 50/50%, 60/40%, 70/30%, and 80/20%. The manual training samples were selected randomly from ground truth data. We iterated damage mapping with random manual training samples 50 times, and report the median of accuracies in Table 9.

**Table 9.** Comparison of automated training samples with manual ones.

| | Manual Training Samples (Train/Test Ratio %) | | | | Automated Training Samples |
|---|---|---|---|---|---|
| | **50/50** | **60/40** | **70/30** | **80/20** | |
| OA (%) | 64 | 65.85 | 65.57 | 60 | 82 |
| KC (%) | 49 | 50.86 | 51.36 | 40.79 | 74.01 |

According to Table 9, it is evident that, compared to the manual selection of training samples, the automatic selection of samples has dramatically improved the overall accuracy and kappa coefficient of damage mapping by over 22% and 33%, respectively. This result reflects the considerable ability of the designed strict rules in opting for more discriminant samples which result in more inter-class variances over the four mentioned damage degrees. As a result, the automated training samples scenario outperforms manual training data with all train/test ratios. Essentially, data annotation for prediction is an elaborate and time-consuming part of supervised classification, particularly in large-scale studies. Addressing this issue would be more vital when it comes to time-sensitive actions such as damage assessment. This issue will even be intensified if more class numbers are added, which makes self-supervised damage level detection more intricate. Another point is the high variance within classes caused by the wide variety of possible damage to buildings, making damage mapping more cumbersome. That is why we test and validate the capability of auto-training samples for damage mapping in this section. In conclusion, the results in this section clarify that our suggested method for generating the damage map after an earthquake, based on auto-training samples and fusing deep and non-deep features using the OECAE network, is time-efficient, accurate, and cost-effective. It is for this reason that the proposed approach in this study could have a large contribution to bridging the gap of applying remote-sensing-based damage detection for timely hazard management and rescue processes.

## 5. Conclusions and Future Research

Practically speaking, there are few studies that have investigated post-earthquake building damage detection. From an operational perspective, a damage mapping method should be efficient in terms of time, accuracy, and cost. Our proposed method boosted the automation of damage detection through the automatic selection of training samples, with an overall accuracy of 93.07%, using physical characteristics of damage degrees with respect to the reputable EMS-98 damage standard. Furthermore, we designed a one-epoch autoencoder network that showed promising results in prompting feature extraction. All in all, we generally explored the effect of the following factors on building damage mapping after an earthquake: (1) feature type, (2) learning method, and (3) method used to select training samples (by the user or automatically). Our results verified that the fusion of non-deep and deep features can significantly enhance the accuracy of the damage map. The remarkable near-accuracy of simple machine learning methods, such as SVM, to deep MLP algorithms was also revealed, both with an overall accuracy of 82%. For these reasons, our solution allows for more actionable damage mapping by following three steps: (1) automatic selection of required training samples, (2) feature extraction by synergic use of deep and non-deep features, and (3) detecting damage level by means of a simple classification algorithm such as SVM or MLP. Similar to many previous related studies, we conducted an accuracy assessment based on legendary accuracy indices. As these indices are merely mathematical, novel adapted accuracy indices should be developed in future studies for the practical evaluation of different damage mapping methods. The probability of various forms of damage occurring to buildings, itself being associated with the natural complexity of earthquake damage, leaves abundant space for further progress in this field. To begin with, further studies with more focus on tackling destructive objects in damage detection—especially trees, chimneys, and shadows—need to be undertaken for more accurate mapping. In addition to this, since the inner extent of building polygons was analyzed for damage detection, a damaged building may appear with any damage sign on the orthophoto; thus, it would be beneficial to use heaps of debris around individual buildings in the post-processing of damage maps. Last but not least, it would be a great idea to apply and evaluate the proposed self-supervised methodology to other geographical locations as well as boost the designed rules for automated training samples selection if needed.

**Author Contributions:** Conceptualization, N.T., A.M. and B.S.; methodology, N.T. and A.M; software, N.T.; validation, N.T. and A.M.; formal analysis, N.T. and A.M.; investigation, N.T and A.M.; resources, N.T. and A.M.; data curation, N.T. and A.M.; writing—original draft preparation, N.T.; writing—review and editing, N.T., A.M. and B.S.; visualization, N.T.; supervision, A.M.; project administration, A.M.; funding acquisition, A.M. All authors have read and agreed to the published version of the manuscript.

**Funding:** This research received no external funding.

**Data Availability Statement:** Not applicable.

**Acknowledgments:** The authors would like to thank "Laser Scanners" research laboratory of KNTU university for providing UAV-derived orthophoto and DSM (https://www.researchgate.net/lab/Laser-Scanners-Ali-Mohammadzadeh, accessed on 6 October 2022).

**Conflicts of Interest:** The authors declare no conflict of interest.

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
