# Peer review of "A Rapid Self-Supervised Deep-Learning-Based Method for Post-Earthquake Damage Detection Using UAV Data (Case Study: Sarpol-e Zahab, Iran)"

_remotesensing, doi:10.3390/rs15010123_

Round 1

Reviewer 1 Report

The scientific field of post-earthquake disaster management is of great interest. Towards this direction, this paper refers to the detecting of post-earthquake buildings’ damage using UAV data.

This reviewer believes that the current version of the manuscript is not yet ready for publication; the authors are encouraged to consider the following comments and suggestions and revise the manuscript accordingly.

The EU Copernicus service is not mentioned anywhere in the introduction (literature review) https://emergency.copernicus.eu/mapping/ems/what-copernicus.

One of the key innovations of the current manuscript lies in the word “rapid”. As the authors mention in line 69 “time, accuracy, and cost are three crucial elements that cannot be overlooked” and lines 70-71 “the processing time of damage mapping has been reported in little research up to now”. However, the detailed time it will take to implement the proposed methodology is not mentioned anywhere, to support the term “rapid”.

Furthermore, the research subject Sarpol-e Zahab in Kermanshah, Iran does not have proper justification either. Unless clearly defined, the manuscript should be revised only as a Case Study paper. Authors are encouraged to state that the assumptions they make about the study area, such as in lines 331-332, can be overridden in another study area. It would be very useful to test the proposed methodology in another study area to demonstrate that it could be used elsewhere.

In the proposed methodology, too many algorithms are used and ultimately their usefulness is not clearly presented. It is suggested that the conclusions incorporate all the results of the experiments, i.e. what work flow (methodology) will be followed if there is a case of earthquake disaster.

This reviewer would believe it would be a quality manuscript once the authors properly address the comments above.

Reviewer 2 Report

All in all a sound piece of work with provided examples. What I am having a bit of trouble with is your speed comparisons for using LiDAR to produce a surface model. You used too high of a density for LiDAR (page 10, line 403) and needed a lot more time to produce output. This is unacceptable and should be redone with a normal point cloud density of 10 points per square meter to make it more realistic. This can be quickly done by thinning the point cloud. 

Due to this, your statement made on page 10, line 406 is not correct and needs to be re-evaluated. 

The claim made on page 10, lines 414-416 is too harsh and should be presented in a better format. The given conclusion does not sound proper or scientifically sound. Revision is necessary. 

When describing the use of ENVI LiDAR for the production of nDSM what are the settings used? I do not see a clear representation of settings. 

Graphics need to increase resolution. Some of them are relatively low res. 

Generally, an interesting paper needs work on expression and form. The LiDAR comparison should be re-evaluated and redesigned.  All in all, acceptable. 

Reviewer 3 Report

This paper addresses a hot topic and is interesting. The model is presented logically, and the results are appropriately discussed. There are, however, some concerns that need to be addressed.

Although the abstract is comprehensive, it is quite long. Perhaps it should be revised to be a bit more punchy.

In the keywords, please use the full name of the UAV.

It is very difficult for readers to follow the introduction because it is very long. There is also a need to revise its logic. My suggestion is to divide the introduction into two sections along with the literature review. Additionally, the aim and objective of the paper should be clearly stated.

More information in section 3 should be provided.

The provided figures in figure 1 should be presented in a better manner. Moreover, this figure is not adequately explained in the paper's body.

It is necessary to revise the presentation of figure 2.
